# Analysis of *CFTR* Mutation Spectrum in Ethnic Russian Cystic Fibrosis Patients

**DOI:** 10.3390/genes11050554

**Published:** 2020-05-15

**Authors:** Nika V. Petrova, Nataliya Y. Kashirskaya, Tatyana A. Vasilyeva, Elena I. Kondratyeva, Elena K. Zhekaite, Anna Y. Voronkova, Victoria D. Sherman, Varvara A. Galkina, Eugeny K. Ginter, Sergey I. Kutsev, Andrey V. Marakhonov, Rena A. Zinchenko

**Affiliations:** Research Centre for Medical Genetics, Moskvorechje Street, 1, 115478 Moscow, Russia; kashirskayanj@mail.ru (N.Y.K.); vasilyeva_debrie@mail.ru (T.A.V.); elenafpk@mail.ru (E.I.K.); elena_zhekayte@gmail.com (E.K.Z.); voronkova111@yandex.ru (A.Y.V.); tovika@yandex.ru (V.D.S.); vgalka06@rambler.ru (V.A.G.); ekginter@mail.ru (E.K.G.); kutsev@mail.ru (S.I.K.); marakhonov@generesearch.ru (A.V.M.); renazinchenko@mail.ru (R.A.Z.)

**Keywords:** cystic fibrosis, *CFTR* gene, common and new pathogenic variants, ethnic Russian population

## Abstract

The distribution and frequency of the *CFTR* gene mutations vary considerably between countries and ethnic groups. Russians are an East Slavic ethnic groups are native to Eastern Europe. Russians, the most numerous people of the Russian Federation (RF), make about 80% of the population. The aim is to reveal the molecular causes of CF in ethnic Russian patients as comprehensively as possible. The analysis of most common *CFTR* mutations utilized for CF diagnosis in multiethnic RF population accounts for about 83% of all CF-causing mutations in 1384 ethnic Russian patients. Variants c.1521_1523delCTT (F508del), c.54-5940_273+10250del21kb (CFTRdele2,3), c.2012delT (2143delT), c.2052_2053insA (2184insA), and c.3691delT (3821delT) are most typical for CF patients of Russian origin. DNA of 154 CF patients, Russian by origin, in whom at least one mutant allele was not previously identified (164 CF alleles), was analyzed by Sanger sequencing followed by the multiplex ligase-dependent probe amplification (MLPA) method. In addition to the 29 variants identified during the previous test for common mutations, 91 pathogenic *CFTR* variants were also revealed: 29 missense, 19 nonsense, 14 frame shift in/del, 17 splicing, 1 in frame ins, and 11 copy number variations (CNV). Each of the 61 variants was revealed once, and 17 twice. Each of the variants c.1209G>C (E403D), c.2128A>T (K710X), c.3883delA (4015delA), and c.3884_3885insT (4016insT) were detected for three, c.1766+1G>A (1898+1G>A) and c.2834C>T (S945L) for four, c.1766+1G>C (1898+1G>C) and c.(743+1_744-1)_(1584+1_1585-1)dup (CFTRdup6b-10) for five, c.2353C>T (R785X) and c.4004T>C (L1335P) for six, c.3929G>A (W1310X) for seven, c.580-1G>T (712-1G>T for eight, and c.1240_1244delCAAAA (1365del5) for 11 unrelated patients. A comprehensive analysis of *CFTR* mutant alleles with sequencing followed by MLPA, allowed not only the identification of 163 of 164 unknown alleles in our patient sample, but also expansion of the mutation spectrum with novel and additional frequent variants for ethnic Russians.

## 1. Introduction

Cystic fibrosis (CF, OMIM#219700) is an autosomal recessive condition resulting from the pathogenic variants in the CF transmembrane regulator (*CFTR*) gene. CF is a hereditary disease caused by impaired epithelial chloride channel CFTR function. Variants are classified as disease causing, not disease causing, of variable clinical significance, or of unknown clinical significance. More than 2000 different variants of the *CFTR* gene sequence have been revealed, the pathogenicity of 20% of which is established [1,2]. In many populations the most frequent pathogenic variant of the *CFTR* gene (*ABCC7*) is F508del, which accounts for approximately two thirds of all *CFTR* alleles, with a decreasing prevalence from Northwest to Southeast Europe. The remaining third of alleles are substantially heterogeneous, with fewer than 20 mutations occurring at a worldwide frequency of more than 0.1%. Some variants can reach a higher frequency in certain populations, due to a founder effect in religious, ethnic or geographical isolates [3]. The spectrum and frequency of *CFTR* gene sequence variants vary significantly in different countries and ethnic groups, which suggests the development of regional molecular diagnostics protocols to optimize medical and genetic care for CF patients [4].

The diagnosis of CF was proven by typical pulmonary or gastrointestinal symptoms or positive neonatal screening, or the diagnosis of CF in a sibling, as well as at least one of the following: two positive sweat chloride tests, or the identification of two *CFTR* pathologic variants in trans according to the guidelines of the European Cystic Fibrosis Society as well as the Russian National Consensus on Cystic Fibrosis [5,6].

Molecular genetic studies on CF have been conducted in the Laboratory of Genetic Epidemiology of the Research Centre for Medical Genetics for a long period of time starting from the year 1989. To date, the laboratory has analyzed the DNA of more than 3400 CF patients, the clinical diagnosis was confirmed in the Scientific-Clinical Department for Cystic Fibrosis of the Research Centre for Medical Genetics. Thereby, 87.4% of the CF patients we examined live in the European part of Russia. More than 85% are Russian or come from marriages between Russians and persons belonging to other ethnic groups. According to the Russian Registry of cystic fibrosis patients of 2017 (RF CF Registry), among at least 212 pathogenic variants of the *CFTR* gene eleven variants are the most frequent ones in the Russian Federation (their relative frequencies exceed 1% in the sample of tested patients) and they are F508del with a share of 52.81%, CFTRdele2,3—6.21%, E92K—3.00%, 2143delT—2.15%, 3849+10kbC>T—2.02%, W1282X—1.90%, 2184insA—1.85%, 1677delTA—1.81%, N1303K—1.54%, G542X—1.35%, and L138ins with 1.24% [7]. All other *CFTR* variants identified in Russian patients share 12.35%. The frequencies and spectrum of variants of the *CFTR* gene vary in different regions. This is caused by specific ethnic background of the population, as well as by different population processes occurring on different territories inhabited by the same ethnos. Thus, in the North Caucasus Federal District (NCFD), three variants are the most frequent ones: F508del (25.0%), 1677delTA (21.5%), and W1282X (17.2%) [7]. A study of *CFTR* gene variants’ spectra in different NCFD ethnic groups revealed a high proportion of variant W1282X (88%) for Karachays [8], and variants 1677delTA (81.5%) and E92K (12.5%) for Chechens [9]. The most frequent variants in the Volga Federal District (VFD) are F508del (50.5%), E92K (8.7%) and CFTRdele2,3 (5.0%) [7]. A high share of E92K variant in VFD is due to the prevalence of this variant for Chuvash (55%) [10]. The second most frequent variant for Chuvash CF patients is F508del (30%) [9], although this value is lower than in the total sample of CF patients (according to the Registry of CF patients in the Russian Federation 2017, [7]).

Russian East Slavic ethnos is the most numerous people in the Russian Federation (RF) (more than 111,000,000 people), which makes 77.7% of the population of the country according to census of 2010 [11]. In the European part of RF, Russians make 85%–90% of the population.

The aim is to describe the Russian-specific spectrum of pathogenic variants of the *CFTR* gene, testing of which could increase the informativeness of DNA diagnostics in regions with a predominantly Russian population, as well to establish a basis for forming a patient base for possible targeted therapy.

## 2. Materials and Methods 

Initially, *CFTR* genotyping of 1384 CF patients (ethnic Russians) from all-Russian sample (3457 CF patients) tested in the Laboratory of Genetic Epidemiology, Research Centre for Medical Genetics were analyzed. The diagnosis of CF was made in the Scientific-Clinical Department for Cystic Fibrosis, Research Centre for Medical Genetics or in regional CF centers according to the accepted standards [10]. Diagnosis was confirmed by analysis of clinical presentation and Gibson–Cooke sweat test, with chloride ion concentrations of 60 mmol/L or higher defining positive result. The assignment of patients’ Russian ancestry was based on self- or parents’ reports. The study included 154 CF Russian patients, 90% of whom came from the European part of the Russian Federation and 10% from Siberian or Far Eastern regions, for all of them at least one mutant allele was not identified.

Patients or their parents signed an informed consent to the study. The research protocol was approved by the Ethical Committee of Research Centre for Medical Genetics (Research Centre for Medical Genetics, 115522, Moscow, Moskvorechie St., 1, Russian Federation, Protocol No.17/2006 of 02.02.2006).

Molecular diagnostics consists of three consecutive stages.

First stage included analysis of 33 frequent *CFTR* variants (c.54-5940_273+10250del21kb (p.Ser18Argfs*16, CFTRdele2,3), c.254G>A (p.Gly85Glu, G85E), c.262_263delTT (p.Leu88IlefsX22, 394delTT), c.274G>A (p.Glu92Lys, E92K), c.350G>A (p.Arg117His, R117H), c.413_415dupTAC (p.Leu138dup; L138ins), c.472dupA (p.Ser158LysfsX5, 604insA), c.489+1G>T (621+1G>T), c.1000C>T (p.Arg334Trp, R334W), c.1040G>C (p.Arg347Pro, R347P), c.1397C>G (p.Ser466X, Ser466X), c.1519_1521delATC (p.Ile507del, I507del), c.1521_1523delCTT (p.Phe508del, F508del), c.1545_1546delTA (p.Tyr515X, 1677delTA), c.1585-1G>A (1717-1G>A), c.1624G>T (p.Gly542X, G542X), c.1652G>A (p.Gly551Asp, G551D), c.1657C>T (p.Arg553X, R553X), c.2012delT (p.Leu671X, 2143delT), c.2051_2052delAAinsG (p.Lys684SerfsX38, 2183AA>G), c.2052_2053insA (p.Gln685ThrfsX4, 2184insA), c.2657+5G>A (2789+5A>G), c.3140-16T>A (3272-16T>A), c.3476C>T (p.Ser1159Phe, S1159F), c.3475T>C (p.Ser1159Pro; S1159P), c.3535_3536insTCAA (p.Thr1179IlefsX17, 3667ins4), c.3587C>G (p.Ser1196X, S1196X), c.3691delT (p.Ser1231ProfsX4, 3821delT), c.3718-2477C>T (3849+10kbC-T), c.3816_3817delGT (p.Ser1273LeufsX28, 3944delGT), c.3844T>C (p.Trp1282Arg, W1282R), c.3846G>A (p.Trp1282X, W1282X), c.3909C>G (p.Asn1303Lys, N1303K), representing a routine Russian Federation panel that identifies up to 85% of mutant CF alleles as described previously [12].

Second stage included analysis of *CFTR* gene coding sequence, exon-intron junctions and 5′-UTR sequence by Sanger sequencing as described previously [12]. Variant pathogenicity status (only pathogenic or likely pathogenic variants were reported) was established using consensus recommendations of the American College of Medical Genetics and Genomics and the Association for Molecular Pathology for interpretation of sequence variants and Russian recommendations. The frequencies of identified alleles in general populations were established based on the GnomAD browser (https://gnomad.broadinstitute.org/). The predicted functional effect of missense variants was determined through SIFT, FATHMM and Radial SVM prediction algorithms as well as GERP++ and PhyloP conservation scores. Intronic and splicing variants were analyzed using Human Splicing Finder tool v. 2.4.1. Novel variants were submitted to the CFTR2 website dataset (https://cftr2.org/), CFTR1 (http://www.genet.sickkids.on.ca/cftr). Pathogenic variants of the *CFTR* gene are denoted according to the legacy nomenclature, besides novel variants named according to the HGVS nomenclature for NM_000492.4 (*CFTR*) transcript variant. 

Third stage intended to search for large rearrangements in chromosome region 7q31.2 (deletions/duplications–CNV) involved the *CFTR* gene locus by the multiplex ligation-dependent probe amplification (MLPA) method in case when no pathogenic allele was detected or an allele with uncertain significance was identified at the previous stages. MLPA analysis was performed with SALSA MLPA probemix P091-D2 CFTR (MRC-Holland, Amsterdam, the Netherlands) according to the manufacturer’s recommendation. The MLPA results were analyzed using Coffalyser.Net (MRC-Holland) [12]. 

Variants phase was checked by segregation analysis in proband and healthy parents.

The gIVS6a_415_IVS10_987Dup26817bp (CFTRdup6b-10) duplication and its boundaries were previously described by F.M. Hantash and co-authors [13]: The fragments duplicated started 415 bp downstream of exon 6a, in IVS6a, and spanned exons 6b, 7, 8, 9, and 10, breaking at 2987 bp downstream of exon 10 in IVS10. The duplicated region is 26,817 bp. Two pairs of primers have been developed to clarify the boundaries of CFTRdup6b-10 duplications identified in Russian CF patients. One flanks the junction area of rupture points of intron 11 (10 as in the legacy nomenclature) and intron 6a (6): IVS10F 5′-TCAGGAAATGGCAATGGGGT-3′ and IVS6aR 5′-GGCTCTGGTGTGATGATCCATA-3′. A 359 bp fragment from these primers is amplified only from the allele carrying the duplication. The second pair (INT10F 5′-GGGGTTGGGAAGTGATTCCA-3′ and INT10R 5′-GCCATCAGCTAGGCTTCTGTA-3′) flanks the rupture area of the intron 10, amplification occurs only from the normal sequence of the intron 10 of the *CFTR* gene leading to a product of 234 bp. 

To compare variant frequencies, the Fisher test was used. The significance level was considered to be *p* ≤ 0.05.

## 3. Results

When developing a routinely used mutation panel, the laboratory’s own data [14], the results of the first collaborative study [15], and studies of other Russian laboratories (in St. Petersburg [16], Bashkortostan [17], and Tomsk [18]) were considered. The panel includes 33 pathogenic variants of the *CFTR* gene identified in patients from different regions of the Russian Federation, as well as the variants specific for certain ethnic groups [7,8,9,10], and allows identification of up to 85% of mutant alleles in all-Russian population [12]. 

At the first stage, the results of testing 33 pathogenic variants of the *CFTR* gene in DNA of 1384 ethnic Russians with CF (previously performed in the laboratory of genetic epidemiology) were analyzed. Thereby, 29 out of 33 tested variants were revealed (Table 1). In addition to F508del and CFTRdele2,3, eight more variants can be referred to as frequent ones for ethnic Russians (frequency of variants 2143delT, 3849+10kbC-T and 2184insA exceed 2%, variants N1303K, G542X, E92K, W1282X, and L138ins exceed 1%). The mutation detection rate of the used panel of tested variants is 83% in the sample of ethnic Russians (Table 1). In 932 patients, two mutant variants were identified, in 426 patients only one pathogenic variant was detected, both alleles were not detected in 26 patients.

On the second stage, 154 ethnic Russians affected by CF, for whom one or both mutant alleles were not identified when analyzing 33 mutations, were selected from the sample of 1384 CF patients for further analysis. Their genotypes were presented in Appendix A. There was a total of 164 unidentified mutant alleles of the *CFTR* gene. 

As a result, in addition to 29 identified frequent mutations, 91 pathogenic (or likely pathogenic) genetic variants in the *CFTR* gene were detected (Table 2). Of these, 29 are missense mutations, 19 nonsense mutations, 14 frame-shift mutations (11 deletions and three insertions)), 17 splice-site, one in-frame insertion, 11 large rearrangements (eight deletions and three duplications).

## 4. Discussion

At present, routine DNA testing of patients includes analysis for 33 pathogenic variants in the *CFTR* gene (Materials and methods section). The spectrum of variants included in the first stage of molecular genetic research was developed gradually. Therefore, 1384 ethnic Russian CF patients are included in the present study, for whom all 33 variants have been tested.

The choice of the spectrum of mutations for routine analysis is conditioned by the results obtained in the course of our own studies [14], studies conducted in different laboratories of the Russian Federation on independent samples of CF patients [6,12,16,17], and data on the prevalence of *CFTR* gene mutations in a global sample of CF patients published by the World Health Organization [18] and presented in the *CFTR* mutation database CFTR1 [19].

Mutations c.1521_1523delCTT (F508del), c.1624G>T (G542X), c.1652G>A (G551D), c.1657C>T (R553X), c.3846G>A (W1282X), c.3909C>G (N1303K), c.489+1G>T (621+1G>T), c.350G>A (R117H), and c.1585-1G>A (1717-1G>A), are among ones the most common in the world [18,20]. Therefore, first of all these mutations were included in the analysis of Russian patients. Variants c.1519_1521delATC (I507del), c.254G>A (G85E), c.3718-2477C>T (3849+10kbC-T), c.1000C>T (R334W), and c.1040G>C (R347P), although not among the most common in the world, are quite common for many populations with specific ethnic background. In 1993–1995, in order to detect pathogenic variants specific to the Russian population, a joint study of the coding sequence of the *CFTR* gene was carried out with the Institute of Biogenetics (Brest, France) by denaturing gradient gel electrophoresis with subsequent sequencing in a sample of 50 patients. It was shown that, in addition to the previously detected *CFTR* gene mutations, the mutations c.2012delT (2143delT), c.2052_2053insA (2184insA), c.262_263delTT (394delTT), and c.3691delT (3821delT) can also be considered frequent for ethnic Russian CF patients. [15]. In the collaborative study of Dörk T. with co-authors [21], in which our laboratory also participated, the predominant distribution of mutation c.54-5940_273+10250del21kb (CFTRdele2,3) was shown for the populations of Eastern Europe and the relative frequency of this mutation was determined for the studied Russian patients (7.2%); second in frequency after the mutation c.1521_1523delCTT (F508del). T.E. Ivashchenko [16] for the first time describes the variants c.1545_1546delTA (1677delTA), c.3587C>G (S1196X) and c.3844T>C (W1282R), relatively frequent for CF patients from Russia. The variant c.1545_1546delTA (1677delTA) was shown to be common for Georgian patients, whereas variants c.3587C>G (S1196X) and c.3844T>C (W1282R) were identified for Russian CF patients. In a study conducted in our laboratory, variants c.3535_3536insTCAA (3667ins4), c.3816_3817delGT (3944delGT), c.472dupA (604insA), and c.413_415dupTAC (L138ins) were identified and included in the frequent mutations’ panel [14].

### 4.1. Similarity and Difference of Frequency Profiles of Common CF Variants in Two Samples of Russian Patients and the Data of CFTR2

A comparison of frequency profiles of 33 variants tested at the first stage shows similarity of frequency distributions for ethnic Russian patients and for patients of All-Russian sample (Table 1): the most frequent is c.1521_1523delCTT (F508del) (54.99% and 52.81%, respectively), the second in frequency is c.54-5940_273+10250del21kb (CFTRdele2,3) (7.59% and 6.21%), and frequencies of eight more variants exceed 1%. This similarity is not surprising, as ethnic Russians make up the majority (over 85%) of CF patients in the Russian Federation. However, there also are differences. Frequencies of the variants c.1521_1523delCTT (F508del) (*p* = 0.059), c.54-5940_273+10250del21kb (CFTRdele2,3) (*p* = 0.018), c.2012delT (2143delT) (*p* = 0.109), c.3718-2477C>T (3849+10kbC>T), c.2052_2053insA (2184insA), c.3909C>G (N1303K), c.1624G>T (G542X), c.3844T>C (W1282R), c.1397C>G (pSer466X), c.3691delT (3821delT), c.3816_3817delGT (3944delGT) are higher for ethnic Russian patients’ sample than for the all-Russian one (Table 1, Figure 1). Perhaps, this is due to the fact that these variants are typical for ethnic Russians and may reflect this ancestry. While frequencies of other variants prevail in the all-Russian sample, which reflects the fact that these variants prevail among patients belonging to other ethnic groups. Thus, the frequency of variant c.1545_1546delTA (1677delTA) for ethnic Russians is much lower than for the all-Russian sample (0.18% and 1.81%, respectively, *p* < 0.0001). The variant c.1545_1546delTA (1677delTA) is predominantly distributed in the North Caucasus populations (Chechens, Ingush, Kumyks) [9,12]. The frequency of variant c.262_263delTT (394delTT) for ethnic Russians is lower than for the all-Russian sample (0.54% vs. 0.94%, *p* = 0.074, although difference is not significant). In the Russian Federation, it is more often found among the population associated with the past settlement of the Finno-Ugric peoples in northwestern European regions and in the Volga-Ural region [12,17]. The frequency of variant c.274G>A (E92K) for ethnic Russian patients is almost three times less than for the all-Russian sample (1.05% vs. 3.00%, *p* < 0.0001). The frequency of variant c.274G>A (p.Glu92Lys, E92K) is maximum for Chuvash (up to 55%) [10], high for Tatars (6.67%), Bashkirs (6.25%) [7], Chechens (12.5%) [9]. The frequency of c.3846G>A (p.Trp1282X, W1282X) is significant higher in the all-Russian sample (RF CF Registry) than in ethnic Russian patients (1.16% vs. 1.90%, *p* = 0.012).

When comparing CF-causing variant frequencies in ethnic Russian CF patients to the CFTR2 database [22], significant frequency difference was found (Fig. 1, Appendix A). So, the frequencies of c.54-5940_273+10250del21kb (p.Ser18Argfs*16, CFTRdele2,3), c.2012delT (p.Leu671X, 2143delT), c.3718-2477C>T (3849+10kbC-T), c.2052_2053insA (p.Gln685ThrfsX4, 2184insA), c.274G>A (p.Glu92Lys, E92K), c.413_415dupTAC (p.Leu138dup; L138ins) and some other variants appear higher in ethnic Russian patients while the frequencies of c.1521_1523delCTT (p.Phe508del, F508del), c.1624G>T (p.Gly542X, G542X), c.2657+5G>A (2789+5A>G), c.489+1G>T (621+1G>T), c.1657C>T (p.Arg553X, R553X), c.254G>A (p.Gly85Glu, G85E), c.1040G>C (p.Arg347Pro, R347P), c.350G>A (p.Arg117His, R117H) were lower. Variants c.3844T>C (p.Trp1282Arg, W1282R), c.3140-16T>A (3272-16T>A), and c.3816_3817delGT (p.Ser1273LeufsX28, 3944delGT) were not listed in CFTR2. Variants c.1652G>A (p.Gly551Asp, G551D), c.1585-1G>A (1717-1G>A), and c.3476C>T (p.Ser1159Phe, S1159F) were not found in tested cohort of Russian patients (Appendix A). However, the differences in frequencies in these latter series involve rare variants and their significance remains unknown.

### 4.2. Sanger Sequencing Detection of the CFTR Gene Variants

As a result of analysis of the coding sequence and regions of exon-intron junctions 80 variants in addition to preliminary tested common *CFTR* gene variants were identified. 61 variants identified in this work were identified on one chromosome and 17 on two chromosomes (Table 2). Each of the variants c.1209G>C (E403D), c.2128A>T (K710X), c.3883delA (4015delA) and c.3884_3885insT (4016insT) were detected for three, c.1766+1G>A (1898+1G>A) and c.2834C>T (S945L) for four, c.1766+1G>C (1898+1G>C) and c.(743+1_744-1)_(1584+1_1585-1)dup (CFTRdup6b-10) for five, c.2353C>T (R785X) and c.4004T>C (L1335P) for six, c.3929G>A (W1310X) for seven, c.580-1G>T (712-1G>T) for eight, and c.1240_1244delCAAAA (1365del5) for 11 unrelated patients (Table 2).

Some of genetic variants identified in sequencing were first discovered in this study. Description of 15 is presented in a previously published paper [23]. Nine of these variants are nonsense mutations (c.252T>A (p.Tyr84X), c.831G>A (p.Trp277X), c.1083G>A (p.Trp361X), c.3112C>T (p.Gln1038X)) or frame-shift mutations (c.264_268delATATT (p.Leu88PhefsX21), c.1219delG (p.Glu407AsnfsX35), c.1608delA (p.Asp537ThrfsX3), c.1795dupA (p.Thr599AsnfsX2), c.3189delG (p.Trp1063X), resulting in the formation of premature stop codon (Table 1). Variant c.490-1G>C breaks the acceptor site of 5 exon splicing. These variants belong to the category of PVS1 null variants (pathogenic variant sequence) according to the criteria of classification of pathogenicity of genetic variants [24]. Variant c.1792_1793insAAA (p.Lys598dup) leads to the insertion of lysine into position 598, and clinical significance of the variant is assessed as pathogenic. The clinical significance of the missense mutations (c.358G>C (p.Ala120Pro), c.1382G>A (p.Gly461Glu), c.1513A>C (p.Asn505His), c.1525G>C (p.Gly509Arg)) is assessed as probably pathogenic.

Eight more variants are presented for the first time. Two variants-nonsense mutations (c.1204G>T (p.Glu402X), c.2617G>T (p.Glu873X)) and one deletion with frame shift (c.2312delA (p.Asn771ThrfsX2)) are concluded to be PVS1 null variants according to the ACMG classification. Variant c.2989-2A>C is a violation of the 19 exon splicing site. Three missense mutations (c.613C>A (p.Pro205Thr), c.1352G>T (p.Gly451Val), c.1589T>C (p.Ile530Thr), c.3107C>A (p.Thr1036Asn)), the clinical significance of which is assessed as probably pathogenic according to the recommendations [24]. The characteristics of the phenotypes of patients who carry rare missense variants are presented in Appendix A.

### 4.3. CNV in Russian CF Patients Detected by MLPA

Large rearrangements of the *CFTR* gene were found for 18 unrelated patients, which is 10.8% (18/166) of the tested mutant alleles and should account for about 1% in the total sample of all mutant alleles in Russians. The MLPA method revealed 11 large rearrangements of the *CFTR* gene: three duplications and eight deletions (Table 1). Four of the large rearrangements were detected in several families. Thus, the duplication of a fragment covering 7–11 (6b–10) exons was detected for five unrelated patients. The testing system we developed allowed us to confirm that the duplication detected had the same frames as previously described in the literature [13]. In the RF CF Registry 2017, this variant was noted for six more unrelated patients. Thus, CFTRdup6b-10 was detected in eleven unrelated patients. Six of them live in the Volga-Ural region, three in the Central region. It should be noted that two patients from the Volga-Ural region belong to the other ethnic groups: one-Bashkir and one-Udmurt.

Each of the deletions, c.(53+1_54-1)_(164+1_165-1)del (CFTRdele2), c.[(1679-1_1680+1)_(2490+1_2491-1)del[;](2908+1_2989-1)del] (CFTRdele12,13;del16) and c.(273-1_274+1)_(869+1_870-1)del(1209-1_1210+1)_(1392+1_1393+1)del (CFTRdel4-7;del9-10) was detected twice. Complex deletion, CFTRdele12,13;del16, was detected for two patients from unrelated families living in the Moscow region; deletion CFTRdel4-7;del9-10 for two families from the Kaliningrad region and the Republic of Buryatia; deletion CFTRdele2 in families from the Transbaikal region and Irkutsk region.

### 4.4. Detection Rate of Three-Stage Analysis of CFTR Gene in Russian CF Patients

As a result of analysis of the coding sequence and regions of exon-intron junctions and subsequent search for large rearrangements, 163 out of 164 alleles were identified that were not detected after preliminary testing of frequent variants of the *CFTR* gene.

In one patient only variant E217G with the F508del in trans was detected after sequencing and MLPA. In NCBI-ClinVar database variant E217G is considered to be variant of conflicting interpretation of pathogenicity (benign; likely benign; uncertain significance) [25]. In the study by Lee J.H. et al. [26] it was shown that non-synonymous E217G mutation in the M470 background caused a 60%–80% reduction in *CFTR*-dependent Cl^−^ currents and HCO_3_^−^ transport activities. So we might suggest that the clinical presentation in that patient is due to complex allele E217G-M470 (Appendix A).

The second mutant allele of the *CFTR* gene could not be identified in one sample. Failure to identify the second pathologic mutation in the *CFTR* gene after sequencing the coding sequence and searching for large rearrangements may be due to the location of the pathogenic variant either in inner regions of the introns, or in regulatory regions of the *CFTR* gene, or in regulatory regions outside the *CFTR* gene. Indeed, such variants have been recently identified, for example, c.1680-883A>G, c.2989-313A>T, c.3469-1304C>G, or c.3874-4522A>G, that lead to the creation of a new donor splice site and the activation of a cryptic acceptor splice site, resulting in the inclusion of an additional pseudo-exon (PE) and the loss of wild type (WT) CFTR transcripts [27].

## 5. Conclusions

In a representative sample of CF patients (ethnic Russians), the spectrum of 33 routinely analyzed (in Russia) variants of the *CFTR* gene was studied. It was shown that, out of 29 identified variants, frequencies of only 10 exceed 1%, and the mutation detection rate of testing did not exceed 85%. Consistent use of sequencing and MLPA methods has allowed us to identify a significant variety of *CFTR* gene mutations spectrum (91 additional genetic variants), to expand the spectrum of frequent variants (c.1766+1G>C (1898+1G>C), c.2353C>T (R785X), c.(743+1_744-1)_(1584+1_1585-1)dup (CFTRdup6b-10), c.4004T>C (L1335P), c.3929G>A (W1310X), c.580-1G>T (712-1G>T), c.1240_1244delCAAAA (1365del5), detected for five and more unrelated patients, to increase the detection rate of identified mutant alleles for Russian CF patients up to 99.4%, consistently using the strategy of Sanger sequencing and MLPA analysis. This information can be useful for the further optimization of medical genetic counseling in CF high-risk families, for improving the neonatal screening program for CF, and for making decision about the possible CFTR modulators therapy in the future. The identification of previously unknown CF-pathogenic or likely-pathogenic variants is a useful piece of information for diagnostic testing not only in Russia, but worldwide, and can be considered as a contribution to the general knowledge about the *CFTR* variant heterogeneity.

## Figures and Tables

**Figure 1 genes-11-00554-f001:**
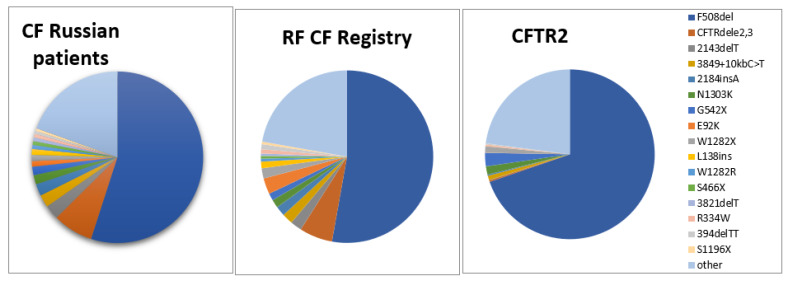
Frequency distribution of *CFTR* gene variants in three samples of CF patients: ethnic Russians, CF RF Registry, CFTR2 database [22].

**Table 1 genes-11-00554-t001:** Frequencies of 33 variants of *CFTR* gene in a sample of 1384 ethnic Russians and in a nationwide sample of CF patients (RF CF Registry) [7].

No.	Variants	Number	%	% in RF CF Registry
1	c.1521_1523delCTT (p.Phe508del, F508del)	1522	54.99	52.81
2	c.54-5940_273+10250del21kb (p.Ser18Argfs*16, CFTRdele2,3)	210	7.59	6.21
3	c.2012delT (p.Leu671X, 2143delT)	75	2.71	2.15
4	c.3718-2477C>T (3849+10kbC-T)	65	2.35	2.02
5	c.2052_2053insA (p.Gln685ThrfsX4, 2184insA)	62	2.24	1.85
6	c.3909C>G (p.Asn1303Lys, N1303K)	48	1.73	1.54
7	c.1624G>T (p.Gly542X, G542X)	44	1.59	1.35
8	c.274G>A (p.Glu92Lys, E92K)	29	1.05	3.00
9	c.3846G>A (p.Trp1282X, W1282X)	32	1.16	1.90
10	c.413_415dupTAC (p.Leu138dup; L138ins)	31	1.12	1.24
11	c.3844T>C (p.Trp1282Arg, W1282R)	21	0.76	0.55
12	c.1397C>G (p.Ser466X, Ser466X)	20	0.72	0.50
13	c.3691delT (p.Ser1231ProfsX4, 3821delT)	19	0.69	0.46
14	c.1000C>T (p.Arg334Trp, R334W)	19	0.69	0.80
15	c.262_263delTT (p.Leu88IlefsX22, 394delTT)	15	0.54	0.94
16	c.3587C>G (p.Ser1196X, S1196X)	14	0.51	0.48
17	c.3816_3817delGT (p.Ser1273LeufsX28, 3944delGT)	12	0.43	0.27
18	c.2657+5G>A (2789+5A>G)	10	0.36	0.48
19	c.489+1G>T (621+1G>T)	7	0.25	0.18
20	c.3140-16T>A (3272-16T>A)	6	0.22	0.34
21	c.1657C>T (p.Arg553X, R553X)	5	0.18	0.18
22	c.1545_1546delTA (p.Tyr515X, 1677delTA)	5	0.18	1.81
23	c.3535_3536insTCAA (p.Thr1179IlefsX17, 3667ins4)	4	0.14	0.10
24	c.254G>A (p.Gly85Glu, G85E)	4	0.14	0.10
25	c.472dupA (p.Ser158LysfsX5, 604insA)	3	0.11	0.10
26	c.2051_2052delAAinsG (p.Lys684SerfsX38, 2183AA>G)	3	0.11	0.04
27	c.3475T>C (p.Ser1159Pro; S1159P)	3	0.11	0.10
28	c.1040G>C (p.Arg347Pro, R347P)	2	0.07	0.10
29	c.350G>A (p.Arg117His, R117H)	1	0.04	0.04
30	c.1519_1521delATC (p.Ile507del, I507del)	0	-	0
31	c.1585-1G>A (1717-1G>A)	0	-	0.04
32	c.1652G>A (p.Gly551Asp, G551D)	0	-	0.04
33	c.3476C>T (p.Ser1159Phe, S1159F)	0	-	0.11
	Identified	2290	82.78	
	Not identified	478	17.22	
	**Total**	2768		

**Table 2 genes-11-00554-t002:** The *CFTR* gene variants additionally identified in 154 previously screened Russian patients.

No.	Variant According to cDNA	Protein Change	Legacy Name	Exon/Intron ^1^	Number	Mutation Type
1	c.43delC	p.Leu15PhefsX1	175delC	1e	2	sd
2	c.53+1G>T		185+1G->T	1i	2	s
3	c.79G>T	p.Gly27X	G27X	2e	1	n
4	c.115C>T	p.Gln39X	Q39X	2e	1	n
5	c.223C>T	p.Arg75X	R75X	3e	1	n
6	c.252T>A	p.Tyr84X		3e	2	n
7	c.264_268delATATT	p.Leu88PhefsX21		3e	1	sd
8	c.274-6T>C		406-6T>C	3i	1	s
9	c.274G>T	p.Glu92X	E92X	4e	1	n
10	c.293A>G	p.Gln98Arg	Q98R	4e	1	m
11	c.358G>C	p.Ala120Pro		4e	1	m
12	c.422C>A	p.Ala141Asp	A141D	4e	1	m
13	c.490-1G>C			4i	1	s
14	c.580-1G>T		712-1G->T	5i	8	s
15	c.613C>A	p.Pro205Thr		6a e	1	m
16	c.650A>G	p.Glu217Gly	E217G	6a e	1	m
17	c.831G>A	p.Trp277X		6b e	1	n
18	c.940G>A	p.Gly314Arg	G314R	7e	1	m
19	c.[1075C>A;1079C>A]	p.[Gln359Lys;Thr360Lys]	Q359K/T360K	7e	1	m
20	c.1083G>A	p.Trp361X		7e	2	n
21	c.1086T>A	p.Tyr362X	Y362X	7e	1	n
22	c.1204G>T	p.Glu402X		8e	1	n
23	c.1209G>C	p.Glu403Asp	E403D	8e	3	m
24	c.[1210−12[5];1210-34TG[12]]		5T;TG12	7i	1	s
25	c.1219delG	p.Glu407AsnfsX35		9e	1	sd
26	c.1352G>T	p.Gly451Val		9e	1	m
27	c.1240_1244delCAAAA	p.Asn415X	1365del5	9e	11	sd
28	c.1364C>A	p.Ala455Glu	A455E	9e	1	m
29	c.1382G>A	p.Gly461Glu		9e	1	m
30	c.1438G>T	p.Gly480Cys	G480C	10e	1	m
31	c.1501A>G	p.Thr501Ala	T501A	10e	1	m
32	c.1513A>C	p.Asn505His		10e	1	m
33	c.1525G>C	p.Gly509Arg		10e	1	m
34	c.1528delG	p.Val510PhefsX17	1660delG	10e	1	sd
35	c.1584+1G>A		1716+1G>A	10i	1	s
36	c.1589T>C	p.Ile530Thr		11e	1	m
37	c.1608delA	p.Asp537ThrfsX3		11e	2	sd
38	c.1646G>A	p.Ser549Asn	S549N	11e	1	m
39	c.1705T>C	p.Tyr569His	Y569H	12e	1	m
40	c.1735G>T	p.Asp579Tyr	D579Y	12e	2	m
41	c.1766+2T>C			12i	2	s
42	c.1766+1G>A		1898+1G>A	12i	4	s
43	c.1766+1G>C		1898+1G>C	12i	5	s
44	c.1792_1793insAAA	p.Lys598dup	K598ins	13e	1	i
45	c.1795dupA	p.Thr599AsnfsX2		13e	1	si
46	c.1911delG	p.Gln637HisfsX26	2043delG	13e	2	sd
47	c.2128A>T	p.Lys710X	K710X	13e	3	n
48	c.2195T>G	p.Leu732X	L732X	13e	1	n
49	c.2290C>T	p.Arg764X	R764X	13e	1	n
50	c.2312delA	p.Asn771ThrfsX2		13e	1	sd
51	c.2353C>T	p.Arg785X	R785X	13e	6	n
52	c.2374C>T	p.Arg792X	R792X	13e	1	n
53	c.2417A>G	p.Asp806Gly	D806G	13e	1	m
54	c.2589_2599delAATTTGGTGCT	p.Ile864SerfsX28	2721del11	14a e	2	sd
55	c.2617G>T	p.Glu873X		14a e	1	n
56	c.2658-2A>G		2790-2A->G	14b i	1	s
57	c.2780T>C	p.Leu927Pro	L927P	15e	1	m
58	c.2834C>T	p.Ser945Leu	S945L	15e	4	m
59	c.2909G>A	p.Gly970Asp	G970D	16e	1	m
60	c.2936A>T	p.Asp979Val	D979V	16e	1	m
61	c.2988+1G>A		3120+1G->A	16i	1	s
62	c.2989-2A>C			16i	1	s
63	c.2989-2A>G		3121-2A->G	16i	1	s
64	c.3107C>A	p.Thr1036Asn		17a e	1	m
65	c.3112C>T	p.Gln1038X		17a e	1	n
66	c.3189delG	p.Trp1063X		17b e	1	n
67	c.3472C>T	p.Arg1158X	R1158X	19e	2	n
68	c.3484C>T	p.Arg1162X	R1162X	19e	2	n
69	c.3528delC	p.Lys1177SerfsX15	3659delC	19e	2	sd
70	c.3763T>C	p.Ser1255Pro	S1255P	20e	2	m
71	c.3775A>T	p.Arg1259X		20e	1	n
72	c.3872A>G	p.Gln1291Arg	Q1291R	20e	1	m
73	c.3874-2A>G		4006-2A->G	20i	1	s
74	c.3883delA	p.Ile1295PhefsX33	4015delA	21e	3	sd
75	c.3884_3885insT	p.Ser1297PhefsX5	4016insT	21e	3	si
76	c.3929G>A	p.Trp1310X	W1310X	21e	7	n
77	c.3963+1G>T		4095+1G->T	21i	1	s
78	c.4004T>C	p.Leu1335Pro	L1335P	22e	6	m
79	c.4242+1G>A		4374+G->A	23i	1	s
80	c.4296_4297insGA	p.Ser1435GlyfsX14	4428insGA	24e	2	si
81	c.(?-1)_(1584+1_1585-1)del		CFTRdele1-10		1	CNV
82	c.(53+2-54-1)_(273+1_274-1)del		CFTRdele2,3(non 21kb)1		1	CNV
83	c.(273+1_274-1)_(743+1_744-1)del		CFTRdele4-6a		1	CNV
84	c.(273-1_274+1)_(869+1_870-1)del(1209-1_1210+1)_(1392+1_1393+1)del		CFTRdel4-7;del9-10		2	CNV
85	c.(489+1_490-1)_(1392+1_1393-1)del		CFTRdele5-10		1	CNV
86	c.(53+1_54-1)_(164+1_165+1)del		CFTRdele21		2	CNV
87	c.(53+1_54-1)_(869+1_870+1)del		CFTRdele2-7		1	CNV
88	c.(1679+1_1680-1)_(2490+1_2491-1)del(2908+1_2909-1)del		CFTRdele12,13;del161		2	CNV
89	c.(743+1_744-1)_(1584+1_1585-1)dup		CFTRdup6b-10 (gIVS6a+415_IVS10+2987Dup26817bp)		5	CNV
90	c.(743-1_744+1)_(869+1_870-1)dup		CFTRdup6b,7		1	CNV
91	c.(4136+1_4137-1)_(*1_?)dup		CFTRdup23,24		1	CNV

Note: (^1^)–exon numbering according to legacy nomenclature, (m)-missense mutation, (n)–nonsense mutation, (sd)–frame-shift deletion, (si)-frame-shift insertion, (s)-splice-site, (i)–in-frame insertion, (CNV)–copy number variation (large rearrangement).

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
