# Peer review of "Analysis of CFTR Mutation Spectrum in Ethnic Russian Cystic Fibrosis Patients"

_genes, 2020, doi:10.3390/genes11050554_

Round 1

Reviewer 1 Report

The manuscript by Petrova at al reports the most common CFTR genotypes in patients of diverse Russian ethnic groups with Cystic Fibrosis disease from Russian Federation. They point out that certain pathogenic variants of CFTR allele more and others less frequent in Russian ethnic groups compare to Russian nationwide CF Registry. This variance based on the geographic distribution of ethnic groups. They analyzed CFTR gene variants in 1384 ethnic Russian CF patients using a genetic panel of 33 that was fitted to identify the most common Russian CF mutations as a first stage test. They used MLPA method combined with Sanger sequencing, as a second stage test, to reveal further undetected mutations in 154 ethnic Russian CF patients. The combination of these two methods detected 99.4% mutant alleles in Russian patients with Cystic Fibrosis. I recommend publication of the manuscript with minor modifications. Comments are as below:

  • I would suggest using a pie or bar chart to visualize and compare the most frequent mutation in ethnic Russian and All Russian CF population besides the tables. You may consider comparing it with the CFTR2 database too.
  • I suggest elaborating on the discussion on how could this study improve the future diagnostic recommendation and methodology of genetic testing. Do the authors recommend decreasing or increasing the first stage panel regarding ethnic group or geographic region? Should the first stage panel be fitted to ethnic groups?
  • How could influence the therapeutic decision (modulator therapy) of the newly revealed ultra-rare CF variants for the affected patients with Cystic Fibrosis?

Minor modification:

  • In line 228: 11 patients described with the c.1240_1244delCAAAA mutation and only 10 listed for The geographical regions.

Author Response

We would like to thank the reviewers for their careful reading of the manuscript and thoughtful comments. We hope that the additions that we made will be sufficient to address the reviewers concerns. Please see below our responses to each specific reviewer comment.

Reviewer 1

The manuscript by Petrova at al reports the most common CFTR genotypes in patients of diverse Russian ethnic groups with Cystic Fibrosis disease from Russian Federation. They point out that certain pathogenic variants of CFTR allele more and others less frequent in Russian ethnic groups compare to Russian nationwide CF Registry. This variance based on the geographic distribution of ethnic groups. They analyzed CFTR gene variants in 1384 ethnic Russian CF patients using a genetic panel of 33 that was fitted to identify the most common Russian CF mutations as a first stage test. They used MLPA method combined with Sanger sequencing, as a second stage test, to reveal further undetected mutations in 154 ethnic Russian CF patients. The combination of these two methods detected 99.4% mutant alleles in Russian patients with Cystic Fibrosis. I recommend publication of the manuscript with minor modifications. Comments are as below:

I would suggest using a pie or bar chart to visualize and compare the most frequent mutation in ethnic Russian and All Russian CF population besides the tables. You may consider comparing it with the CFTR2 database too.

               Response: We have added the pie chart with comparison of frequencies. The comparison also is now in Supplementary Material.

I suggest elaborating on the discussion on how could this study improve the future diagnostic recommendation and methodology of genetic testing. Do the authors recommend decreasing or increasing the first stage panel regarding ethnic group or geographic region? Should the first stage panel be fitted to ethnic groups?

               Response: We added “This information can  be useful for further optimization of medical genetic counseling in CF high-risk families, for improving the neonatal screening program for CF and for making decision about the  possible CFTR modulators  therapy in the future. The identification of previously unknown CF-pathogenic or likely-pathogenic variants is a useful information for diagnostic testing not only in Russia but worldwide and can be considered as a contribution to the general knowledge about the CFTR variants heterogeneity. “ to Conclusion  

How could influence the therapeutic decision (modulator therapy) of the newly revealed ultra-rare CF variants for the affected patients with Cystic Fibrosis?

               Response: We have added this statement into the Conclusion.

Minor modification:

In line 228: 11 patients described with the c.1240_1244delCAAAA mutation and only 10 listed for The geographical regions.

               Response: We have 11 patients with this variant but we have removed clinical information from the text..

Reviewer 2 Report

The aim of this paper is to characterize undefined mutant alleles in an ethnic population. Currently, only 85% of CF-causing alleles are identified by screens in the Russian Federation. Here the authors screen for an additional 92 mutations in a population with high sweat chloride and one identified CF-causing allele, and captured nearly all CF-causing second alleles in this population. For these patients, identity of the second allele may make them eligible for future clinical trials and treatments.

Strengths: Many CF treatments are designed for specific CFTR mutations classes. These data may make these patients eligible for mutation-class specific treatments. The manuscript highlights the importance of testing additional alleles when most common alleles are not detected.

Weaknesses: There are comparisons made without the use of statistics. While this manuscript illustrates the strengths above, and potentially life-altering to these patients, there are no new findings or interpretations. In addition to thorough text editing, shortening the manuscript to focus on the strengths above may improve it.

Specific comments:

The case summaries in the discussion do not relate to the motivation of the paper. Could be summarized in a table.

Line 35 chlorine instead of chloride is used

In abstract and line 92, spell out MLPA on first use

Line 125 29 variants

Table 1, last column, should read “% according to registry”

Table 2, last column, difficult to tell what this is according to the heading. Mutation type have its own heading

Line 250, chronic airway damage of P.aeruginosa… how do you know the damage was made by P.aeruoginosa?

Line 174 needs a reference

Author Response

We would like to thank the reviewers for their careful reading of the manuscript and thoughtful comments. We hope that the additions that we made will be sufficient to address the reviewers concerns. Please see below our responses to each specific reviewer comment.

Reviewer 2

The aim of this paper is to characterize undefined mutant alleles in an ethnic population. Currently, only 85% of CF-causing alleles are identified by screens in the Russian Federation. Here the authors screen for an additional 92 mutations in a population with high sweat chloride and one identified CF-causing allele, and captured nearly all CF-causing second alleles in this population. For these patients, identity of the second allele may make them eligible for future clinical trials and treatments.

Strengths: Many CF treatments are designed for specific CFTR mutations classes. These data may make these patients eligible for mutation-class specific treatments. The manuscript highlights the importance of testing additional alleles when most common alleles are not detected.

Weaknesses: There are comparisons made without the use of statistics. While this manuscript illustrates the strengths above, and potentially life-altering to these patients, there are no new findings or interpretations. In addition to thorough text editing, shortening the manuscript to focus on the strengths above may improve it.

               Response: We have shortened the manuscript and removed all clinical information as well as placed the description of novel variants in Supplementary Material. We used Fisher test for comparison of Russian, All-Russian and CFTR database in the Text and Supplementary Materials.

Specific comments:

The case summaries in the discussion do not relate to the motivation of the paper. Could be summarized in a table.

               Response: We removed clinical description of two mutations (1367del5, 721-1G>T), and made the Supplementary Table 3 with clinical features of rare missense variants.

Line 35 chlorine instead of chloride is used –

               Response: Corrected.

In abstract and line 92, spell out MLPA on first use -

               Response: Corrected.

Line 125 29 variants - 

               Response: Corrected. We meant 29 out of 33 tested variants.

Table 1, last column, should read “% according to registry”

               Response: Corrected.

Table 2, last column, difficult to tell what this is according to the heading. Mutation type have its own heading

               Response: table is amended

Line 250, chronic airway damage of P.aeruginosa… how do you know the damage was made by P.aeruginosa?  

               Response: We mean “chronic P. aeruginosa infection of the respiratory tract’. Updated.

Line 174 needs a reference  

               Response: Updated.

Reviewer 3 Report

The manuscript entitled 'Analysis of CFTR mutation spectrum in ethnic Russian CF patients' by NV Petrova et al describes the analysis of the CFTR gene in 1384 cystic fibrosis patients of Russian ethnicity with a three-phased strategy. In the first phase, a selected group of 33 CF-causing variants compiled from earlier series of Russian Federation cases were tested. Twenty-nine of the 33 variants were identified accounting for 2291 of the 2768 alleles. (Note there appears to be a discrepancy in the actual numbers of alleles listed in Table 1.)

For the second phase, 154 individuals with neither or only one identified CF-causing allele (in the first phase, for a total of 164 unknown alleles), were analyzed using Sanger sequencing of amplified exon, exon-intron boundary and regulatory regions. A final analysis phase then involved the MLPA method to identify large deletions or duplications, and direct assessment of the CFTRdup6b-10 variant.  

The manuscript lists identified CFTR variants and frequencies in individuals of Russian ancestry with CF; it is entirely descriptive in nature. The frequencies of common variants in the presented group generally confirm those previously established, and some new variants were identified. Ethnic origin and new variants are of value for molecular diagnosis of cystic fibrosis.  

There are a number of aspects of the presentation that should be improved, see Specific comments to Authors. Further, the manuscript text would benefit from grammatical editing and clarification. The Minor comments listed are not complete, but include the most obvious typos or words that confound understanding. 

Specific comments: 

1.    There are several numbering inconsistencies throughout the manuscript, leading to confusion in interpretation.  For examples, the summed totals for the third column of Table 1 and the last column of Table 2, appear inconsistent with the Table content.  

2.     It is initially unclear if the 1384 CF cases had contributed to previously indicated studies, or if this set is entirely new and distinct.  Was there overlap with previous sets that were mentioned? Relevant information in this regard does appear in the first three paragraphs of the Discussion section (lines 155–192) but this should be presented earlier, as the content appears to relate entirely to information known prior to the described investigations.

3.    The assignment of Russian ancestry appears to be based on self-reporting, this should be stated, or at least clarified.

4.     The three versus two phases (as indicated in the manuscript) should be clarified, as it appeared that not all 154 individuals investigated in the second phase were subjected to MPLA analysis. Is this the correct interpretation?

5.     The Materials and Methods section is not complete. Specifically, resources for the Sanger sequencing or the MPLA methods were not given, or referenced. Applied software, including that used for assignment of pathogenicity were also not listed or referenced.

6.     The text and Table 2 refer to 91 pathogenic variants but only 90 are described on lines 145–147, page 4?

7.     There are a number of additional issues with Table 2, the title is indicated as being identical to Table 1? Also, the text describes that 163/164 alleles were identified, but it appears that only 161 are listed? The columns could be more effectively spaced to maximize presentation clarity. The column titles should be clarified – for example, the last column indicates 'amount'. (?, a more appropriate heading may be 'Number and allele type'). The legacy names for allele numbers 81-91 do not appear to be correct, they include the character '1', that may have been intended to be a footnote?  It would appear that the footnote reference could appear in the fourth column heading.

8.     The CF-causing variants found in the 144 individuals with only single CF-causing alleles could in indicated and incorporated with Table 1 such that they would not need to be listed on lines 135–140, page 4. 

9.     Beyond the comments in point 2 above, the Discussion section is still very long, and could be shortened. The use of subsections would further improve its presentation. 

10.   While there were trends regarding the frequency of CF-causing variants between the ethnic Russians and the All-Russian (or Russian Federation) samples, some of the differences for the lower frequency variants may not be conclusive as they involve low numbers, first full paragraph on page 8. This limitation should be indicated.

11.   The numbers of variants described on lines 218–224, page 8, do not add up to 163 as the text indicates. 

12.   The sentence ‘Variants … composition.’, lines 173-176, page 7 is not clear.

13.   The Discussion text, lines 225–254 is lengthy, and does not appear to provide substantial new information (at least what was critical was not made clear). Further, why are patient phenotype details given for this group of patients, but not specifically for others?  Patient phenotype details would be more interesting for individuals with variants that were not previously described, particularly those with novel missense variants.

14.   Regarding the group of 8 variants that are indicated to be new, page 9, at least three have been reported previously (and are listed at Ref. 19), including c.1204G>T, c.2312delA, and c.2989-2A>C (the latter in a patient of Japanese descent).  Also, note, 'Three missense variants' are indicated, but four are listed on lines 270 and 271, page 9.

15.   The Conclusions section implies that 99.4% of alleles can be 'consistently' identified in patients of Russian origin, however this reflected only a single test group. It should be clarified what the 'detection rate' refers to here. In addition, the identification of previously unknown CF-pathogenic or likely-pathogenic variants should also be noted as an important conclusion.

16.   Have or will the authors share their findings with public databases listing CFTR variants, this is useful information for diagnostic testing around the world?  

17.   CNVs, CVS and CVs abbreviations are used, include one only with a definition and be consistent throughout.

Additional minor/typo issues:

18.   'CFTRdele2.3' should read 'CFTRdele2,3', line 17, page 1. It appears that this variant is also referred to as CFTR2,3del on line 125, page 3 and in Table 2?

19.   'chlorine' should read 'chloride', line 35, page 1.

20.   'proven' should read 'established', line 38, page 1.

21.   What is 'the combined sample' indicated on line 44, page 1?

22.   'in Russian patients variants' should read 'variants in Russian patients', line 47, page 2.

23.   'routinely used in the Russian Federation … mutant alleles' should read 'a routine Russian Federation panel that identifies up to 85% of mutant alleles', lines 87 and 88, page 2.

24.   'is about 83%' should read 'is 83%', line 128, page 3.

25.   'or both (10) mutant alleles' should read 'or both (20) mutant alleles', line 133, page 4.

26.   'carrying variant … in compound' should read 'with the N1303K variant on the second allele', line 226, and 'bases … registered' should read 'databases, this variant was not listed', line 227, page 8.

27.   The use of the division symbol is confusing on lines 232–233, page 8.  An n-dash may be more appropriate to indicate range.

28.   'Chronic sowing of' (?) could read 'chronic infection with', lines 236 and 251, on pages 8 and 9, respectively.

29.   'second frequent' should read 'second-most frequent', line 240, page 8.

30.   Is the 'Russian Register of CF Patients 2017' the same as 2017-2018, line 42, page 1 or 'Russian Register of CF Patients 2016-2017', line 280 page 9?

31.   'are treated as' should read 'are concluded to be', line 269, page 9.

32.   Define 'PS1 null variant', line 269, page 9?

33.   'Pussia', on line 366, page 11 should read 'Russia'?

Author Response

We would like to thank the reviewers for their careful reading of the manuscript and thoughtful comments. We hope that the additions that we made will be sufficient to address the reviewers concerns. Please see below our responses to each specific reviewer comment.

Reviewer 3

The manuscript entitled 'Analysis of CFTR mutation spectrum in ethnic Russian CF patients' by NV Petrova et al describes the analysis of the CFTR gene in 1384 cystic fibrosis patients of Russian ethnicity with a three-phased strategy. In the first phase, a selected group of 33 CF-causing variants compiled from earlier series of Russian Federation cases were tested. Twenty-nine of the 33 variants were identified accounting for 2291 of the 2768 alleles. (Note there appears to be a discrepancy in the actual numbers of alleles listed in Table 1.)

For the second phase, 154 individuals with neither or only one identified CF-causing allele (in the first phase, for a total of 164 unknown alleles), were analyzed using Sanger sequencing of amplified exon, exon-intron boundary and regulatory regions. A final analysis phase then involved the MLPA method to identify large deletions or duplications, and direct assessment of the CFTRdup6b-10 variant. 

The manuscript lists identified CFTR variants and frequencies in individuals of Russian ancestry with CF; it is entirely descriptive in nature. The frequencies of common variants in the presented group generally confirm those previously established, and some new variants were identified. Ethnic origin and new variants are of value for molecular diagnosis of cystic fibrosis. 

There are a number of aspects of the presentation that should be improved, see Specific comments to Authors. Further, the manuscript text would benefit from grammatical editing and clarification. The Minor comments listed are not complete, but include the most obvious typos or words that confound understanding.

Specific comments:

  1. There are several numbering inconsistencies throughout the manuscript, leading to confusion in interpretation. For examples, the summed totals for the third column of Table 1 and the last column of Table 2, appear inconsistent with the Table content.  

               Response: We corrected data in Tables 1 (proportions of variants) and 2 (addition of the seventh column – Mutation type) and throughout the text.

  1. It is initially unclear if the 1384 CF cases had contributed to previously indicated studies, or if this set is entirely new and distinct. Was there overlap with previous sets that were mentioned? Relevant information in this regard does appear in the first three paragraphs of the Discussion section (lines 155–192) but this should be presented earlier, as the content appears to relate entirely to information known prior to the described investigations.

               Response: We now present this information in materials and Methods section.

  1. The assignment of Russian ancestry appears to be based on self-reporting, this should be stated, or at least clarified.

               Response: We changed to “The assignment of patient’s Russian ancestry was based on self- or parents’ reporting”

  1. The three versus two phases (as indicated in the manuscript) should be clarified, as it appeared that not all 154 individuals investigated in the second phase were subjected to MPLA analysis. Is this the correct interpretation?

               Response: We make the Materials and methods Section clearer.

  1. The Materials and Methods section is not complete. Specifically, resources for the Sanger sequencing or the MPLA methods were not given, or referenced. Applied software, including that used for assignment of pathogenicity were also not listed or referenced.

               Response: We make the Materials and methods Section more detailed and added the references.

  1. The text and Table 2 refer to 91 pathogenic variants but only 90 are described on lines 145–147, page 4?

               Response: We missed one missense variant– 29 instead of 28

  1. There are a number of additional issues with Table 2, the title is indicated as being identical to Table 1? Also, the text describes that 163/164 alleles were identified, but it appears that only 161 are listed? The columns could be more effectively spaced to maximize presentation clarity. The column titles should be clarified – for example, the last column indicates 'amount'. (?, a more appropriate heading may be 'Number and allele type'). The legacy names for allele numbers 81-91 do not appear to be correct, they include the character '1', that may have been intended to be a footnote? It would appear that the footnote reference could appear in the fourth column heading.

               Response: The title of Table 2 was corrected to ‘The CFTR gene variants additionally identified in 154 previously screened Russian patients’. The number of identified alleles was 163: we corrected line 37 – c.1608delA – 2 instead 1, line 57 – R785X – 6 instead 5 in Table 2. We change the name of th3d column Table 3 to ‘Number’ and the name of the 4th column Table 3 to ‘Mutation type’. We agree the character '1' should be places to the fourth column heading.

  1. The CF-causing variants found in the 144 individuals with only single CF-causing alleles could in indicated and incorporated with Table 1 such that they would not need to be listed on lines 135–140, page 4.

               Response: We removed the information about 154 patients in Supplement 2. We believe that adding information on the genotypes of 154 patients selected for further study will complicate Table 1. We agree to present this information in a Supplementary table 2. Table 1 presents frequencies; we believe it is important to specify the genotypes of the patients selected for further analysis; the distribution of frequencies in this group may differ from the general group of Russians and the all-Russian sample, albeit due to random reasons: patients were selected randomly, but the condition was one or both unidentified mutations.

  1. Beyond the comments in point 2 above, the Discussion section is still very long, and could be shortened. The use of subsections would further improve its presentation.

               Response: We have divided the Discussion section into subsections and removed the clinical description.

  1. While there were trends regarding the frequency of CF-causing variants between the ethnic Russians and the All-Russian (or Russian Federation) samples, some of the differences for the lower frequency variants may not be conclusive as they involve low numbers, first full paragraph on page 8. This limitation should be indicated.

               Response:  We compared the frequencies of variants in ethnic Russian sample, all-Russian sample and CFTR2 data using Fisher test in Fig. 1 and Supplementary Table 1. We add “Although for some rare variants, the difference in frequency is difficult to prove as they involve low numbers.” in Discussion section.

  1. The numbers of variants described on lines 218–224, page 8, do not add up to 163 as the text indicates.

               Response:  We corrected “61 (instead of 63) variants identified in this work were identified on one chromosome and 17 on two chromosomes (Table 2). Each of the variants c.1209G>C (E403D), c.2128A>T (K710X), c.3883delA (4015delA) and c.3884_3885insT (4016insT) detected for 3, c.1766+1G>A (1898+1G>A) and c.2834C>T (S945L) – for 4, c.1766+1G>C (1898+1G>C) and c.(743+1_744-1)_(1584+1_1585-1)dup (CFTRdup6b-10) – for 5, c.2353C>T (R785X) ( instead of 5) and c.4004T>C (L1335P) – for 6, c.3929G>A (W1310X) – for 7, c.580-1G>T (712-1G>T) – for 8, c.1240_1244delCAAAA (1365del5) for 11 unrelated patients (Table 2).   Thus, the sum of identified alleles is 163.

  1. The sentence ‘Variants … composition.’, lines 173-176, page 7 is not clear.

               Response: We meant that Rare variants could be frequent in certain populations of specific ethnic background.

  1. The Discussion text, lines 225–254 is lengthy, and does not appear to provide substantial new information (at least what was critical was not made clear). Further, why are patient phenotype details given for this group of patients, but not specifically for others? Patient phenotype details would be more interesting for individuals with variants that were not previously described, particularly those with novel missense variants.

               Response: We moved information about clinical presentation of two variants, and give description of rare novel missense mutations in Supplement 3.

  1. Regarding the group of 8 variants that are indicated to be new, page 9, at least three have been reported previously (and are listed at Ref. 19), including c.1204G>T, c.2312delA, and c.2989-2A>C (the latter in a patient of Japanese descent). Also, note, 'Three missense variants' are indicated, but four are listed on lines 270 and 271, page 9.

               Response: We reported variants were submitted by the authors of the current manuscript: variants c.1204G>T, c.2312delA in CFTR1. c.2989-2A>G and c.2989-2A>T are listed in CFTR1 variants, but not c.2989-2A>C. http://www.genet.sickkids.on.ca/cftr/MutationDetailPage.external?sp=421 A patient of Japanese origin have c.2989-2A>G, but not c.2989-2A>C variant. 

  1. The Conclusions section implies that 99.4% of alleles can be 'consistently' identified in patients of Russian origin, however this reflected only a single test group. It should be clarified what the 'detection rate' refers to here. In addition, the identification of previously unknown CF-pathogenic or likely-pathogenic variants should also be noted as an important conclusion.

               Response: We believe that since the patient selection was random (the only condition is that one or both mutations were not identified), the results can be extrapolated to a nationwide sample of CF patients (mutation rate - the proportion of detectable variants using a three-step analysis). We also included in the Conclusion "the identification of previously unknown CF-pathogenic or likely-pathogenic variants should be important"

  1. Have or will the authors share their findings with public databases listing CFTR variants, this is useful information for diagnostic testing around the world?

               Response: We have partially entered our data into the CFTR1 database and plan to do so in the future. In addition, our doctors participate in the annual filling in of the ECFS Patient Registry (https://www.ecfs.eu/ecfspr), and our data are transmitted to the Russian and European Registries.

  1. CNVs, CVS and CVs abbreviations are used, include one only with a definition and be consistent throughout.

               Response: everywhere fixed with CNV.  

Additional minor/typo issues:

  1. 'CFTRdele2.3' should read 'CFTRdele2,3', line 17, page 1. It appears that this variant is also referred to as CFTR2,3del on line 125, page 3 and in Table 2?

               Response: correct name was edited to 'CFTRdele2,3’. In Table 2 - one mutation included deletion of exons 2, 3 according to MLPA analysis, but had other boundaries: it was not detected at PCR with specific primers flanking deletion 2,3(21kb).

  1. 'chlorine' should read 'chloride', line 35, page 1.

               Response: fixed.

  1. 'proven' should read 'established', line 38, page 1.

               Response: fixed.

  1. What is 'the combined sample' indicated on line 44, page 1?

               Response: We meant ‘all-national sample of CF patients’, removed ‘combined’ – line 44 (48).

  1. 'in Russian patients variants' should read 'variants in Russian patients', line 47, page 2.

               Response: fixed.

  1. 'routinely used in the Russian Federation … mutant alleles' should read 'a routine Russian Federation panel that identifies up to 85% of mutant alleles', lines 87 and 88, page 2.

               Response: fixed. Changed to “a routine Russian Federation panel that identifies up to 85% of mutant CF alleles”

  1. 'is about 83%' should read 'is 83%', line 128, page 3.

               Response: fixed.

  1. 'or both (10) mutant alleles' should read 'or both (20) mutant alleles', line 133, page 4.

               Response: fixed. When we wrote 'or both (10) mutant alleles', we meant 10 patients, not alleles. Numbers deleted. – ‘one or both mutant alleles’

  1. 'carrying variant … in compound' should read 'with the N1303K variant on the second allele', line 226, and 'bases … registered' should read 'databases, this variant was not listed', line 227, page 8.

               Response: fixed.

  1. The use of the division symbol is confusing on lines 232–233, page 8. An n-dash may be more appropriate to indicate range.

               Response: fixed. We changed “÷” to “−“ at all lines.

  1. 'Chronic sowing of' (?) could read 'chronic infection with', lines 236 and 251, on pages 8 and 9, respectively.

               Response: fixed.

  1. 'second frequent' should read 'second-most frequent', line 240, page 8.

               Response: fixed.

  1. Is the 'Russian Register of CF Patients 2017' the same as 2017-2018, line 42, page 1 or 'Russian Register of CF Patients 2016-2017', line 280 page 9?

               Response: fixed.

  1. 'are treated as' should read 'are concluded to be', line 269, page 9.

               Response: fixed.

  1. Define 'PS1 null variant', line 269, page 9?

               Response: fixed.

  1. 'Pussia', on line 366, page 11 should read 'Russia'?

               Response: fixed.

Reviewer 4 Report

The manuscript “Analysis of CFTR mutation in ethnic Russian patients” present extensive and interesting data regarding CFTR mutations, which can help to manage patients and to precise genetic counseling. Overall this is a well-executed genetic study on a large number of patients with the identification of several poorly known or totally unknown variants. However, I think that the data are not fully exploited in the current presentation, notably for the rare missense variants. In order to appreciate the disease-liability of a variant in the context of a recessive disease, readers need to know the complete genotype of the patient. Considering the large number of patients it may be difficult to describe everyone but this could be done at least for missense variants or variant identified only once or twice.

  • Important background data are missing in the introduction part, for example: the recessive inheritance of CF, the classification of CFTR variant pathogenicity with some pathogenic variant not being associated with CF but with mild CFTR-related disorders or the fact that CF is caused by the presence of two CF-causing mutations on both alleles. On the other hand, some data regarding the prevalence of certain mutation in specific ethic groups are repeated in the discussion section. The impact of variant identification and interpretation for the patients and their families could also be added in the introduction.
  • Authors should verify the manuscript regarding variant nomenclature for example line 74: p.Ser18Arg*fsX16, line 83: 3237-16T>A is probably 3272-16T>A, table 2 c.264-268delATATT is probably c.264_268delATATT or line 180: p2184insA ; and CFTR should be written in italics when referring to the gene.
  • M&M: It is unclear if the 1384 CF patients were included in a previous publication, as well as the 154 patients with 1 or 2 unknown alleles. Are the 154 patients parts of the 1384 group?
  • Results: line 115 to 122 could be included in the introduction part and are repeated in discussion.
  • In phase 1, the proportion of patients with 2, 1 and 0 mutation identified after testing for 33 mutations should be indicated.
  • The R117H variant (table 1) can be associated in cis with either T5 or T7 allele in IVS8 which modified its pathogenicity: R117H;T7 is by far the most frequent and is considered as a moderate CFTR-RD variant whereas R117H;T5 may be associated with CF. Can the author indicate if the patient in table 1 is R117H;T7 or R117H;T5 ?
  • Legend of table 2 is the same as for table 1.
  • Many of the variants described in table 2 are poorly known or totally unknown, which is the great interest of the present work but the current presentation. It would add a considerable value to the study if the genotypes of the patients and if possible a minimum of clinical data (sweat test, pancreatic status) were present, at least for patients with a rare missense variant or in supplementary material.
  • Was the phase of the variants determined by sequencing the parents of the 154 patients ?
  • Which variant was identified alone ?
  • To my knowledge, E217G (table 2) is not associated with CF and is rather considered as a neutral variant. Its allele frequency is around 1% in East Asian population and 3,5% in Finnish and it has been reported in trans of a CF-causing variant in asymptomatic individuals in the CFTR-France database (https://cftr.iurc.montp.inserm.fr/cgi-bin/affiche.cgi?variant=c.650A%3EG). If the authors consider E217G as a CF-causing variant they should explain why in the text.
  • Line 260 : the presence of a premature stop codon does not necessarily lead to the synthesis of a shortened protein because of nonsense mediated decay. Most of the time mRNA with premature stop codon are degraded and not translated into protein.
  • Line 267 : Three missense mutations (c.613C>A (p.Pro205Thr), c.1352G>T (p.Gly451Val), 270 c.1589T>C (p.Ile530Thr), c.3107C>A (p.Thr1036Asn)) è this make 4 mutations
  • I am not a native English speaker, so I don’t feel qualified to judge the style of the authors, but I am not sure about certain word or sentences for example : chlorine channel (line 35, I have always read chloride channel), “The total share of other identified in Russian patients variants is 12.35% (line 47) or “That is, a total of CFTRdup6b-10 was detected for eleven patients” (line 280) may not be correctly formulated. Authors may need to check manuscript again for correct wording.

Author Response

We would like to thank the reviewers for their careful reading of the manuscript and thoughtful comments. We hope that the additions that we made will be sufficient to address the reviewers concerns. Please see below our responses to each specific reviewer comment.

Reviewer 4

The manuscript “Analysis of CFTR mutation in ethnic Russian patients” present extensive and interesting data regarding CFTR mutations, which can help to manage patients and to precise genetic counseling. Overall this is a well-executed genetic study on a large number of patients with the identification of several poorly known or totally unknown variants. However, I think that the data are not fully exploited in the current presentation, notably for the rare missense variants. In order to appreciate the disease-liability of a variant in the context of a recessive disease, readers need to know the complete genotype of the patient. Considering the large number of patients it may be difficult to describe everyone but this could be done at least for missense variants or variant identified only once or twice.

Important background data are missing in the introduction part, for example: the recessive inheritance of CF, the classification of CFTR variant pathogenicity with some pathogenic variant not being associated with CF but with mild CFTR-related disorders or the fact that CF is caused by the presence of two CF-causing mutations on both alleles. On the other hand, some data regarding the prevalence of certain mutation in specific ethic groups are repeated in the discussion section. The impact of variant identification and interpretation for the patients and their families could also be added in the introduction.

               Response: We updated the introduction.

Authors should verify the manuscript regarding variant nomenclature for example line 74: p.Ser18Arg*fsX16, line 83: 3272-16T>A is probably 3272-16T>A, table 2 c.264-268delATATT is probably c.264_268delATATT or line 180: p2184insA ; and CFTR should be written in italics when referring to the gene.

               Response: We checked the variants names and fixed them if needed.

M&M: It is unclear if the 1384 CF patients were included in a previous publication, as well as the 154 patients with 1 or 2 unknown alleles. Are the 154 patients parts of the 1384 group?

               Response: 1384 CF patients were not included in a previous publication, the 154 patients were the part of the 1384 group. Some of 154 patients were included in  ref.4. Petrova, N.V., Marakhonov, A.V.; Vasilyeva, T.A.; Kashirskaya, N.Y.; Ginter, E.K.; Kutsev, S.I.; Zinchenko, R.A. Comprehensive genotyping reveals novel CFTR variants in cystic fibrosis patients from the Russian Federation. Clin Genet, 2018, 95, 1-4. DOI: 10.1111/cge.13477

Results: line 115 to 122 could be included in the introduction part and are repeated in discussion.

               Response: We have restructured these parts accordingly.

In phase 1, the proportion of patients with 2, 1 and 0 mutation identified after testing for 33 mutations should be indicated.

               Response: We have added “In 932 patients two mutant variants were identified, in 426 patients only one pathogenic variant was detected, both alleles were not detected in 26 patients’

The R117H variant (table 1) can be associated in cis with either T5 or T7 allele in IVS8 which modified its pathogenicity: R117H;T7 is by far the most frequent and is considered as a moderate CFTR-RD variant whereas R117H;T5 may be associated with CF. Can the author indicate if the patient in table 1 is R117H;T7 or R117H;T5 ?

               Response: Patient in table 1 had R117H;T7, but the diagnosis CF was made by our doctors and not canceled, despite the result of genotyping.

Legend of table 2 is the same as for table 1.

               Response: We have corrected the title and the legend “The CFTR gene variants additionally identified in 154 previously screened Russian patients”

Many of the variants described in table 2 are poorly known or totally unknown, which is the great interest of the present work but the current presentation. It would add a considerable value to the study if the genotypes of the patients and if possible a minimum of clinical data (sweat test, pancreatic status) were present, at least for patients with a rare missense variant or in supplementary material.

               Response: We represented this information in Supplement 3.

Was the phase of the variants determined by sequencing the parents of the 154 patients ?

               Response: Linkage phases was proved by family analysis. We added this information to the M&M section.

Which variant was identified alone ?

               Response: The second allele could not been identified in patients with c.264_268delATATT variant.

To my knowledge, E217G (table 2) is not associated with CF and is rather considered as a neutral variant. Its allele frequency is around 1% in East Asian population and 3,5% in Finnish and it has been reported in trans of a CF-causing variant in asymptomatic individuals in the CFTR-France database (https://cftr.iurc.montp.inserm.fr/cgi-bin/affiche.cgi?variant=c.650A%3EG). If the authors consider E217G as a CF-causing variant they should explain why in the text.

               Response: In one patient only variant E217G with the F508del in trans was detected after sequencing and MLPA. In NCBI-ClinVar database variant E217G is considered to be variant of conflicting interpretation of pathogenicity (benign;likely benign;uncertain significance) [22]. In the study by Lee J.H. et al. [25] it was shown that non-synonymous E217G mutation in the M470 background caused a 60–80% reduction in CFTR-dependent Cl  currents and HCO3  transport activities. So we might suggest that the clinical presentation in that patient is due to complex allele E217G-M470 (Supplementary table 3).

Line 260 : the presence of a premature stop codon does not necessarily lead to the synthesis of a shortened protein because of nonsense mediated decay. Most of the time mRNA with premature stop codon are degraded and not translated into protein.

               Response: We agree and removed this sentence

Line 267 : Three missense mutations (c.613C>A (p.Pro205Thr), c.1352G>T (p.Gly451Val), 270 c.1589T>C (p.Ile530Thr), c.3107C>A (p.Thr1036Asn)) è this make 4 mutations

               Response: Yes, four mutations

I am not a native English speaker, so I don’t feel qualified to judge the style of the authors, but I am not sure about certain word or sentences for example : chlorine channel (line 35, I have always read chloride channel), “The total share of other identified in Russian patients variants is 12.35% (line 47) or “That is, a total of CFTRdup6b-10 was detected for eleven patients” (line 280) may not be correctly formulated. Authors may need to check manuscript again for correct wording.

               Response: fixed.

Reviewer 5 Report

Petrova et al describes the distribution and frequency of the CFTR gene mutations in ethnic Russian patients. The aim of the authors is to obtain better tools for CF diagnosis in multiethnic Russian Federation population, for whom the standard analysis accounts for only 83% of all CF-causing mutations.

The manuscript is very interesting, since it describes a very peculiar spectrum of mutations as compared to other countries. Although mainly descriptive, this study will be very useful to improve CFTR diagnostics for ethnic Russian patients.

The authors extensively discuss their results in the light of previous studies. Interestingly, they also include a genotype-phenotype correlation, although very preliminary. 

The only weakness of this study is that it requires professional editing of English language.

Author Response

We would like to thank the reviewers for their careful reading of the manuscript and thoughtful comments. We hope that the additions that we made will be sufficient to address the reviewers concerns. Please see below our responses to each specific reviewer comment.

Reviewer 5

Petrova et al describes the distribution and frequency of the CFTR gene mutations in ethnic Russian patients. The aim of the authors is to obtain better tools for CF diagnosis in multiethnic Russian Federation population, for whom the standard analysis accounts for only 83% of all CF-causing mutations.

The manuscript is very interesting, since it describes a very peculiar spectrum of mutations as compared to other countries. Although mainly descriptive, this study will be very useful to improve CFTR diagnostics for ethnic Russian patients.

The authors extensively discuss their results in the light of previous studies. Interestingly, they also include a genotype-phenotype correlation, although very preliminary.

The only weakness of this study is that it requires professional editing of English language.

               Response: We have substantially edited te manuscript.

Round 2

Reviewer 2 Report

Thank you for putting in considerable time and effort into incorporating changes to the manuscript. The manuscript overall reads better and is more focused than the previous version. I have only minor comments.

Define CNV as copy number variation in abstract

Line 46 - There are 2 periods

Line 50 – Define where. Maybe say “Regional diagnosis of CF varies and may be based on a consistent phenotype or…”. Another sentencing clearly stating RFs approach to diagnosis would be informative here.

Line 86 – Omit “In the beginning”

Table 2 column headings are mis-aligned

Author Response

First of all, we would like to thank the reviewer for their careful reading of the manuscript and thoughtful comments. We hope that the manuscript became better with their suggestions. Please see below our responses to each specific reviewer comment.

Reviewer’s comments

Define CNV as copy number variation in abstract

               Response: We have written out this abbreviation.

Line 46 - There are 2 periods

               Response: We have removed redundant period.

Line 50 – Define where. Maybe say “Regional diagnosis of CF varies and may be based on a consistent phenotype or…”. Another sentencing clearly stating RFs approach to diagnosis would be informative here.

               Response: We have rewritten this paragraph (lines 50-54 in clean copy).

Line 86 – Omit “In the beginning”

               Response: Done.

Table 2 column headings are mis-aligned

               Response: We tried to make the table 2 more readable by adding hyphens into the headings and alignment.